# Global learning opportunities within social innovation in health (GLOWS): A modified Delphi process to identify and pilot core competencies for learning

Emily Wallace[1,2]*, Yusha Tao[3,4], Ogechukwu B. Aribodor[5,6], Zixuan Zhu[3], Angelica Borbón[7], Beatrice Halpaap[8], Bertha M. Chakhame[9], Eunice C. Jacob[5], Fatema Ahmed[10], Joel Msafiri Francis[11], Komang G. Septiawan[12], Kovey Mawuli[13], Linet Mutisya[14], Marlita Putri Ekasari[15], Nwadiuto Okwuniru Azugo[6], Tina Fourie[16], Adriana S. Ruiz[17], Jackeline Alger[18,19], Abigail Ruth Mier[20], Weiming Tang[21], Gloria Aidoo-Frimpong[22], Jackline Nanono[23], Jesson James A. Montealto[13,24], Obidimma Ezezika[25,26], Per Kåks[27], Wenjie Shan[28], Jana Deborah Mier-Alpano[18], Gifty Marley[29], Elizabeth Chen[21], Joseph D. Tucker[1,3]*

1 London School of Hygiene and Tropical Medicine, London, United Kingdom, 2 The Spark Innovation Programme, Health Service Executive, Dublin, Ireland, 3 University of North Carolina Project-China, Guangzhou, China, 4 Dermatology Hospital of South Medical University, Guangzhou, China, 5 Department of Zoology, Nnamdi Azikiwe University, Awka, Nigeria, 6 Social Innovation in Health Initiative (SIHI) Nigeria Hub, Nnamdi Azikiwe University, Awka, Nigeria, 7 Innovation in Public Health, Department of Research in Public Health, National Institute of Health, Bogotá, Colombia, 8 TDR, UNICEF/UNDP/World Bank/WHO Special Programme for Research and Training in Tropical Diseases, Geneva, Switzerland, 9 Kamuzu University of Health Sciences, Lilongwe, Malawi, 10 School of Nursing and Rehabilitation, Shandong University, Jinan, China, 11 Department of Family of Medicine and Primary Care, School of Clinical Medicine, Faculty of Health Sciences, University of the Witwatersrand, Johannesburg, South Africa, 12 Social Innovation in Health Initiative (SIHI) Indonesia at the Center for Tropical Medicine, Faculty of Medicine, Public Health, and Nursing, Universitas Gadjah Mada, Yogyakarta, Indonesia, 13 Precise Consultancy and Training Service, Abu Dhabi, United Arab Emirates, 14 Uppsala University, SIHI hub, Uppsala, Sweden, 15 Division of Management and Community Pharmacy, Department of Pharmaceutics, Faculty of Pharmacy, Universitas Gadjah Mada, Yogyakarta, Indonesia, 16 Because Stories, Storytelling and Communication Agency, Cape Town, South Africa, 17 Innovation in Public Health, Department of Research in Public Health, National Institute of Health, Bogotá, Colombia, 18 Hospital Escuela, Tegucigalpa, Honduras, 19 Instituto de Enfermedades Infecciosas y Parasitología Antonio Vidal, Tegucigalpa, Honduras, 20 Social Innovation in Health Initiative - Philippines Hub, Manila, Philippines, 21 Department of Health Behavior, Gillings School of Global Public Health, University of North Carolina at Chapel Hill, Chapel Hill, North Carolina, United States of America, 22 Department of Epidemiology and Environmental Health, University at Buffalo, State University of New York, New York, New York, United States of America, 23 Makerere University School of Public Health - Department of Community Health and the SIHI Uganda, Kampala, Uganda, 24 Institute of Health Policy and Development Studies, National Institutes of Health of the University of the Philippines Manila, Manila, Philippines, 25 Global Health & Innovation Lab, Faculty of Health Sciences, Western University, London, Ontario, Canada, 26 African Centre for Innovation & Leadership Development, Abuja, Nigeria, 27 Centre for Health and Sustainability, Department of Women's and Children's Health, Uppsala University, Uppsala, Sweden, 28 Department of International Clinics, Shanghai Children's Medical Center, Shanghai Jiao Tong University School of Medicine, Shanghai, China, 29 University of North Carolina at Chapel Hill - Project China, Chapel Hill, North Carolina, United States of America

* Emily.wallace1@alumni.lshtm.ac.uk (EW), jdtucker@med.unc.edu (JDT)

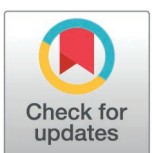

**Data availability statement:** All relevant data are within the manuscript and its Supporting information files.

**Funding:** The author(s) received no specific funding for this work.

**Competing interests:** I have read the journal's policy and the authors of this manuscript have the following competing interests: Angelica Borbón, Bertha M. Chakhame, Adriana S. Ruiz, Abigail Ruth Mier, Obidimma Ezezika, Jana Deborah Mier-Alpano, and Gifty Marley have received honoraria for speaking at the pilot workshops. These payments were provided by SESH, the SIHI Hub in China. The authors declare no other potential conflicts of interest. This does not alter our adherence to PLOS ONE policies on sharing data and materials.

# Abstract

## Background

Social innovation in health refers to the community-engaged process that connects health improvement and social change. The aim of this study was to develop a consensus statement on core learning competencies in social innovation in health and pilot them as part of a participatory training workshop.

## Methods and findings

A modified Delphi Process aggregating data from a scoping review, global open call, and participatory process was organized. Participants were recruited from low, middle, and high-income countries with a range of social innovation experiences. Statements focused on social innovation in health core competencies for learning. Consensus was determined using the RAND/UCLA Appropriateness method. After expressing interest in the project, 68 individuals received the survey. 46 participants completed the first survey, and 35 completed the second. All 28 statements reached consensus, and based on the results of this first survey, some statements were added, amended, and merged to reach 30 consensus statements in the second survey. Competencies were categorized into skills, mindsets, and knowledge. Twenty-five statements had a median Likert rating score of >8 indicating strong agreement. Some competencies reached higher levels of agreement. This included community engagement, which can leverage the collective knowledge and problem-solving abilities of a diverse group of individuals to tackle complex challenges; social entrepreneurship skills including business model knowledge, securing funding, team building, and knowledge of intersectional issues and health inequities. Twelve competencies were then piloted as eight one-hour online workshops, which assessed the feasibility of developing them through online open-access social innovation training sessions. Afterwards,137 participants completed a survey rating their competency on a scale from 1 (not competent) to 5 (very competent),most reported a significant 1-point improvement including in entrepreneurship and understanding intersectionality.

## Conclusion

The results from this study will inform the development of a WHO/TDR conceptual framework which will have implications for training program design and policy.

## Background

Social innovation in health is a community-engaged process that connects health improvement and social change by developing inclusive, context-specific solutions [1]. Rooted in principles of equity, participation, and sustainability, social innovation

has been applied to address complex health challenge, particularly in settings where conventional approaches fall short [2]. By centring local communities to identify and co-create innovative solutions, social innovation not only generates more effective and sustainable outcomes, but also builds local capacity and ownership [3]. Teaching core social innovation competencies can empower communities and other stakeholders to innovate for better health in their localities [4].

Despite its growing application in health systems, particularly in low- and middle-income countries, social innovation remains underrepresented in formal educational frameworks. Training programs often lack clarity around the core competencies required to support effective social innovation in health. While numerous social innovation education programs and training resources exist, content and methods vary significantly and the majority of learning resources have been developed for high-income settings [5–7] This inconsistency presents challenges for scaling and sustaining social innovation globally.

Recognizing this gap, the Special Programme for Research and Training in Tropical Diseases (TDR) and the Social Innovation in Health Initiative (SIHI) have highlighted the need for more social innovation training to catalyse and support the development of novel health solutions and build a pipeline of social innovation researchers [8]. In 2023, TDR (the UNICEF/UNDP/World Bank/WHO Special Programme for Research and Training in Tropical Diseases) and SIHI indicated a need for consensus on social innovation in health learning competencies to be considered in training and further resources.

In response to this critical gap in social innovation in health learning, this study seeks to establish consensus using Delphi methodology on core competencies and piloting them through participatory workshops. Guided by this aim, the study addresses the following questions: 1) What are the core competencies required for effective social innovation in health, 2) To what extent can consensus be achieved among experts on these competencies and 3) Can some of these competencies be taught and engaged with in a virtual workshop setting? The results will inform a conceptual framework which can be adopted in training programmes and resources. Establishing standardized competencies can optimize workforce development especially in resource-constrained settings by aligning education with local health priorities. This can enhance health system responsiveness and support community-driven efforts to reduce health disparities.

## Study context

This study was conducted in collaboration with several institutional partners, each with mandates in global health and social innovation. The Social Innovation in Health Initiative (SIHI) is a global network hosted by various academic and research institutions that supports research and capacity-building to integrate social innovation into health systems, particularly in low- and middle-income countries. Social Entrepreneurship to Spur Health (SESH) is a research group based in China and affiliated with SIHI, which specializes in applying crowdsourcing and participatory methods to improve health outcomes.

## Methods

### Modified Delphi process

To develop consensus, this study used a modified Delphi, an iterative process of consolidating expert opinions into group consensus [9]. It was initially used in the 1950s to predict the impact of technology on society [10]. Modified Delphi approaches have been used across many health fields, especially in areas where there is a lack of evidence on best practices [11]. The RAND/UCLA Appropriateness method is a modified Delphi that involves iterative surveying and then group feedback sessions after the completion of the first survey [12]. This study used a two-round modified Delphi, supplemented by teleconference discussions (Fig 1). Training on the identified competencies was then piloted through online workshops.

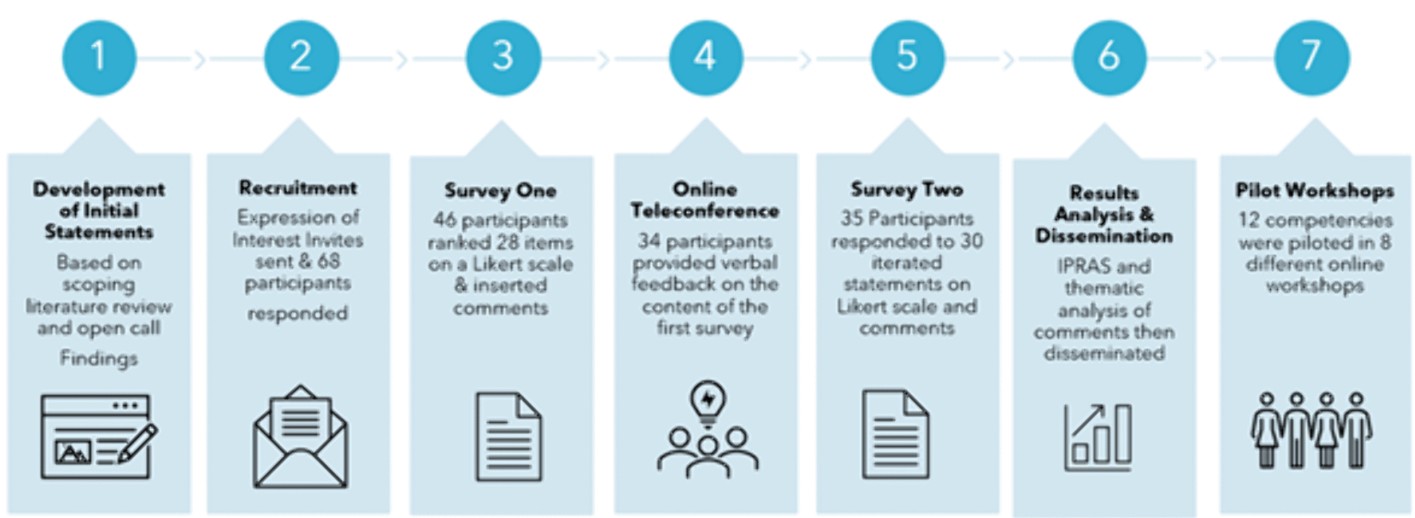

**Fig 1. Steps in Consensus building process.**

The WHO uses consensus methods to develop normative guides for policymakers, national public health agencies, and others [13]. The Modified Delphi Technique offers advantages over other consensus building techniques such as participant anonymity, iterative feedback, and suitability for remote and complex decision-making, making it particularly effective in minimizing groupthink and interpersonal biases as well as enabling diverse international participation. This is in contrast to other consensus processes such as the Nominal Group Technique which is more limited by face-to-face interaction, fewer iterative opportunities, and a higher risk of influence from group dynamics [11].

Ethical approval was received in 2024 from the London School of Hygiene and Tropical Medicine, United Kingdom. There were no potential risks or harms to participants identified as questionnaires did not request sensitive information and participants were not deemed to be a vulnerable population. All participation was voluntary with informed consent. Raw data will be deleted one year after the completion of the study.

## Survey development

The initial development of statements was informed by the results of a scoping literature review and a crowdsourcing open call conducted by Social Entrepreneurship to Spur Health (SESH) and SIHI [14] The open call was a structured crowdsourcing way of soliciting community feedback. Crowdsourcing can be defined as method in which a group of individuals attempt to solve the same problem, then identify exceptional solutions out of the group for implementation. Crowdsourcing open calls invite broad public participation to generate ideas or solutions, often using online platforms [5,15–17].

The crowdsourcing open call received a total of 38 eligible submissions. The scoping literature review included 20 studies. Themes extracted from both the open call and literature review were constructed into statements which were distributed in round one of the modified Delphi process [18]. Statements were categorised into knowledge, skills or attitudes as per the KSA (knowledge, skills, attitudes) framework [19]. The themes identified during the open call and scoping review included mindsets or attitudes, which was further described as items based on empathy building, resilience, and adaptability. The skills-based grouping included communication, advocacy, partnership building, participatory research, and user-centered design, while the knowledge-based competencies contained intersectionality, advocacy, and ethical-related themes (Fig 2) [19].

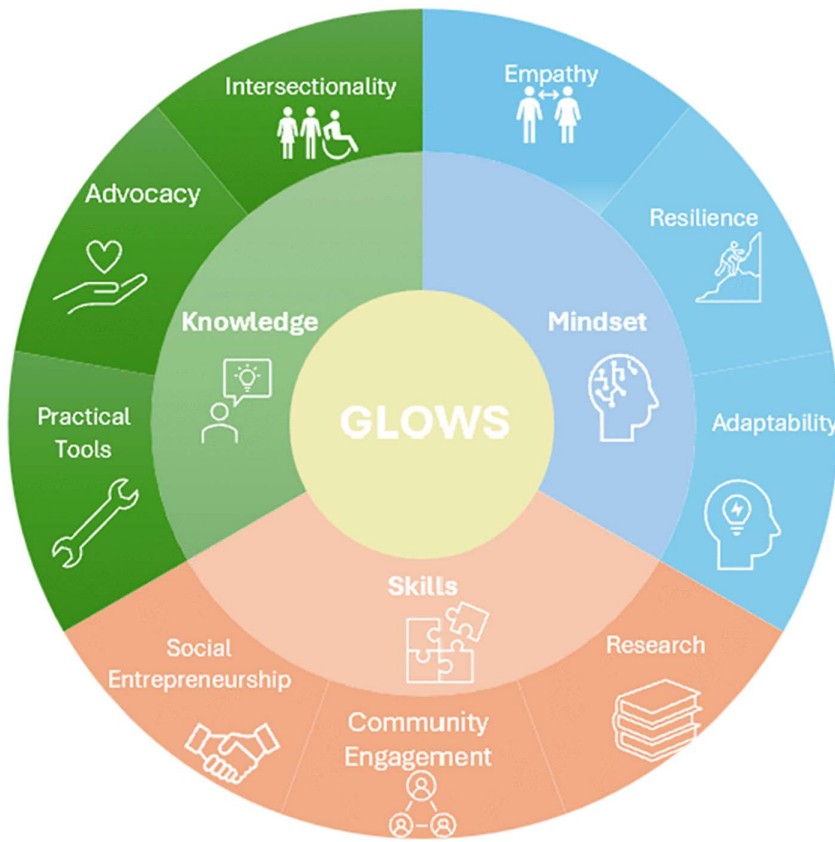

**Fig 2. Core Competencies including Skills, Knowledge, and Mindsets Identified from the Consensus Building Process.**

### Recruitment

A total of 113 members from the SIHI network, corresponding authors from the social innovation in health learning competency scoping review, and academics with a research interest in social innovation in health were invited to the study. Participants were emailed a link to a short survey hosted on the JotForm platform that requested contact details and basic demographic information. Inclusion criteria for participants included the ability to complete questionnaires in English, internet access, and a previous track record in social innovation in health activities. Participation was voluntary, and no incentives were offered.

### Round one

The first survey took approximately ten minutes to complete. Participants had two weeks to complete the survey in early June 2024, and a reminder email was sent five days prior to the deadline. The consent was embedded in the form. Demographic information such as country of birth, education level, gender, and specific details on their experience in social innovation were collected. The questionnaire contained 28 statements on core competencies. These initial competencies were derived from initial results and themes that emerged from a thematic analysis of submissions to the open call and results of the scoping review. Each statement was presented initially with a definition, followed by a statement referring to social innovation in health. The panellists were asked to rank their level of agreement on a scale of 1–9, with 9 corresponding to strongly agree, 7 to agree, 5 to neutral, 3 to disagree, and 1 to strongly disagree. Each statement

was followed by a Likert scale and a comment box where participants could provide feedback and propose statement edits.

## Teleconference session

Participants were then invited to provide verbal feedback on the first survey and discuss statements on Zoom teleconference calls. The session was repeated two different times to facilitate attendance in different time zones. There was discussion on statements that had conflicting comments from the first survey, and then time was allocated for participants to suggest new competencies and propose iterations.

## Round two

Based on analysis of the results of the initial survey, a second survey was developed and distributed via link to JotForm in an email to the participants. The survey contained revised statements and definitions, along with some additional statements which were created based on feedback and comments from participants who completed the first survey and teleconference sessions. The aim of this final online Delphi survey was to assess level of agreement on the iterated statements.

The survey was distributed in June 2024, and participants were given ten days to complete the survey. A reminder email was sent five days prior to the deadline again. Similarly to round one, each statement was followed by both a Likert scale and a comment box where participants could input why the agreed, or disagreed with statements, and proposed edits or additions to statements. Participants had the opportunity to suggest useful open access resources such as publications, videos, websites and courses. They were also asked what their preferred study outputs based on what they think would be the most useful resource in their practice. Finally, they had the opportunity to provide insights on how learning content delivery, strategies and competencies can be tailored to meet the needs of resource limited contexts and low-and-middle-income-countries (LMICs).These questions were included as currently many educational materials are inaccessible to learners in LMICs due to financial and infrastructural barriers. Gathering suggestions for open-access resources can help inform future resource development.

## Analysis

The median, interquartile range, participants' ratings compared to group ratings, and frequencies for each statement were calculated for all statements in both surveys.

The Inter-percentile Range Adjusted for Symmetry (IPRAS), derived from the RAND/UCLA appropriateness method, was used to assess the level of agreement among participants based on their Likert scale responses. IPRAS measures asymmetry in responses across the 9-point Likert scale using a calculation incorporating the inter-percentile range (30th and 70th centile), the median score, and a correction factor for asymmetry. This approach, applied to each rated statement, was developed as an alternative to relying on arbitrary cut-offs [20,21]. The agreement was defined as per RAND definitions and assessed where the median falls [22]. A disagreement index (DI) below 1 indicated consensus, with a median score of 7–9 signifying agreement and 1–3 indicating disagreement. A thematic analysis of the comments was also conducted to identify aspects of statements for iteration and new statement additions to the survey. Themes that were repeatedly highlighted by different participants were included in the next iteration.

After data collection and analysis, all participants received a note of gratitude for their participation as well as feedback with summarised findings from the questionnaire and consensus statement. ACCORD (Accurate Consensus Reporting Document) was used to report study findings [23].

## Pilot online workshop

Twelve core competencies were then piloted as a training workshop series. From Jun 4 through Jul 23, 2024, an online training workshop on social innovation in health (60 minutes per week) was organized by SESH (Social Entrepreneurship

to Spur Health) and SIHI. The workshop series consisted of eight online sessions [24]. Each session was led by a finalist from a global open call, who developed the training content based on one or more of the twelve competencies. To support content quality and relevance, each finalist was paired with a mentor—an experienced academic or practitioner in fields such as social innovation, implementation research, or community engagement—who reviewed the materials and provided feedback throughout the process Workshops covered the following topics: Community engagement, user-centered design and crowdsourcing (week 1), social innovation theories and frameworks and community engagement (week 2), human-centered design for health and empathy building (week 3), intersectional themes such as social determinants of health and health disparities (week 4), co-creation and community engagement (week 5), human-centered design (week 6), leadership and building sustainable innovations (week 7), pitching and storytelling for social innovations (week 8). (S1 Table) Each session included approximately 40 minutes of didactic presentation and 20 minutes of participatory activities, such as live polling, open discussion, and chat-based engagement, tailored to diverse digital contexts and participant bandwidth.

This component of the study was deemed not to be human subject research which was approved by the University of North Carolina Institutional Review Board. Data was de-identified and will be deleted one year after completion of the study.

Baseline assessments gathered information on participants' sociodemographic characteristics and included self-assessment items evaluating their baseline knowledge and skills related to social innovation in health. A self-administered exit survey on the final day evaluated participants' experiences, satisfaction, and knowledge and skills related to social innovation in health by the end of the workshop. Both the baseline and exit surveys included identical items measuring participants' confidence in applying these concepts, using a 5-point Likert scale (1 = Not at all confident to 5 = Extremely confident). The assessment instruments were adapted from previously piloted tools used in similar capacity-building trainings, with contextual refinements for this workshop series.

The Mann-Whitney U test was used to estimate changes by comparing participants self-assessed pre-workshop measures (i.e., reflecting on their knowledge and skills before starting the workshop) with post-workshop measures. All data was analyzed using GraphPad Prism 8.0 statistical software (GraphPad Software Inc., La Jolla, CA, USA).

## Results

A total of 68 individuals expressed interest in participating in the Delphi process in June 2024. Forty-five (66.6%) of those identified as women. Twenty-four countries from all WHO regions were represented. Nigeria, the Philippines, Sweden, and the United States of America had the highest representation from individual countries, with at least five people expressing interest in participating.

### Round one

The response rate for the first survey was 46/68 (67.6%). Participants resided in 18 different countries. Thirty-three (71.7%) were female, and there were various experiences in social innovation and education level, as depicted in Table 1.

All of the 28 statements met the threshold for consensus to varying degrees. A total of 25 of 28 (89.2%) statements had a median Likert rating score of >8 and a Disagreement Index <0.3, indicating strong agreement. Statements that had very strong agreement included communication-based competencies and competencies based on intersectional topics. Other statements, such as manuscript writing for scientific publication, saw diverging perspectives among participants. 28 of 46 (60%) participants entered comments to at least one statement – demonstrating a high level of participant engagement. Based on this commentary, the definitions of 26 statements were elaborated for the second round of the survey. Four statements with common themes were merged, and three new statements were added, including proposed competencies in the area of research skills, navigating regulatory pathways, and definitions and frameworks as a practical tool.

**Table 1. Demographic Details of Participants who completed surveys.**

| Variable | Round 1 (n = 46) | | Round 2 (n = 35) | |
|---|---|---|---|---|
| **WHO Region and Country of Residence** | **n** | **(%)** | **n** | **(%)** |
| **WHO Region of Americas** | **13** | **(28.3)** | **9** | **(25.7)** |
| United States | 9 | (19.6) | 5 | (14.3) |
| Colombia | 2 | (4.3) | 2 | (5.7) |
| Canada | 1 | (2.2) | 1 | (2.9) |
| Honduras | 1 | (2.2) | 1 | (2.9) |
| **WHO African Region** | **11** | **(23.9)** | **9** | **(25.7)** |
| Nigeria | 5 | (10.9) | 4 | (11.4) |
| Malawi | 2 | (4.3) | 1 | (2.9) |
| South Africa | 2 | (4.3) | 1 | (2.9) |
| South Sudan | 1 | (2.2) | 1 | (2.9) |
| Uganda | 1 | (2.2) | 1 | (2.9) |
| Rwanda | 0 | | 1 | (2.9) |
| **Western Pacific** | **11** | **(23.9)** | **8** | **(22.9)** |
| Philippines | 5 | (10.9) | 5 | (14.3) |
| China | 4 | (8.7) | 1 | (2.9) |
| Australia | 2 | (4.3) | 1 | (2.9) |
| Singapore | | | 1 | (2.9) |
| **WHO European Region** | **9** | **(19.6)** | **4** | **(11.4)** |
| Sweden | 5 | (10.9) | 3 | (8.6) |
| United Kingdom | 2 | (4.3) | 1 | (2.9) |
| Switzerland | 1 | (2.2) | 1 | (2.9) |
| Ireland | 1 | (2.2) | 1 | (2.9) |
| **WHO Southeast Asian Region** | **1** | **(2.2)** | **1** | **(2.9)** |
| Indonesia | 1 | (2.2) | 1 | (2.9) |
| **WHO Eastern Mediterranean Region** | **1** | **(2.2)** | **2** | (5.7) |
| Kuwait | 1 | (2.2) | 1 | (2.9) |
| Jordan | 0 | | 1 | (2.9) |
| **Gender** | | | | |
| Female | 33 | (71.7) | 24 | (68.6) |
| Male | 13 | (28.3) | 11 | (31.4) |
| Other | 0 | (0) | 0 | (0) |
| **Education Level** | | | | |
| PhD/MD | 22 | (47.8) | 18 | (51.4) |
| Masters | 16 | (34.8) | 11 | (31.4) |
| Undergraduate | 7 | (15.2) | 5 | (14.3) |
| High School | 1 | (2.2) | 1 | (2.9) |
| **Social Innovation Experience** | | | | |
| Attended social innovation training days as a participant | 23 | (50.0) | 23 | (65.7) |
| Designed/implemented social innovations in health | 26 | (56.5) | 20 | (57.1) |
| Delivered social innovation training/teaching content. | 27 | (58.7) | 18 | (51.4) |
| Analysed social innovations in health | 1 | (2.2) | 0 | (0) |

## Teleconference sessions

A total of 34 participants joined teleconference sessions. The statements and feedback from the first round were discussed. Different methods of learning were also discussed, such as classroom-based teaching styles, self-directed learning, and experiential learning. Participants gave feedback that research skills such as literature appraisal were important to include in the subsequent iteration, along with an emphasis on more participatory approaches to achieve social innovation competencies.

## Round two

The response rate was 36/68 (52.9%) for the second survey. One participant was excluded because they did not have experience in social innovation. Twenty-four (68.6%) participants were women, and there was representation from over 20 countries. The majority of participants had either a PhD, MD or master's degree. Participants had varying experience in social innovation in health among participants, including teaching and learning about social innovation and delivering and implementing innovations. The threshold for consensus was reached in all 30 statements in the second survey as shown in Table 2. In general, comments varied in sentiment, with the majority expressing agreement with the statements.

## Competencies

All 30 statements reached the threshold for consensus. We identified a total of 29 competencies, which corresponded to each statement in the second round of surveys in addition to the preamble which gave context to the project and key definitions. The competency statements were categorized into three groups: 3 related to mindset/attitude, 17 focused on skills, and 9 based on knowledge-related competencies (Fig 2).

The preamble was an added item in the second survey as per feedback on the first edition. It contained the definition of social innovation, outlined the reasons for why training and education are important to achieve impactful innovation in health, the primary users of the consensus statement and different mechanisms to achieve competencies in social innovation. The preamble had a median Likert rating of 8 and 33/35 (94.2%) agreed with the statement. The disagreement index also indicated consensus.

Empathy, adaptability, and resilience were all mindset-based competencies that were included. Each had a median Likert ranking of at least 8, indicating strong agreement, and 34/35 (97.1%) of participants agreed to varying degrees with the statements. While, overall, participants felt that having the right mindset should be included as an aim for learners to enable success in social innovation, numerous participants felt that one's mindset is an intrinsic trait that is difficult to teach. On the other hand, some felt that it is feasible to develop mindsets.

There was also strong agreement for most of the skill-based competencies. All communication-based competencies (such as community engagement, storytelling, and pitching) received high ratings. Storytelling and pitching were identified as necessary to gain support, mobilize resources, and inspire others to action. Community engagement leverages the collective knowledge, creativity, and problem-solving abilities of a diverse group of individuals to tackle complex challenges. Social entrepreneurship skills such as business model knowledge, securing funding, team building, campaigning, and developing partnerships were also deemed to be important for social innovation.

Research skills were another important competency identified. Some research skills such as community based participatory research reached very strong levels of agreement resulting in a DI of 0.13, while others slightly weaker agreement such as manuscript writing for scientific publication had a disagreement index (DI) of 0.65.

Generally, participants indicated that social innovators should accomplish a broad range of skills, but it was acknowledged in many responses that some of these skill sets are not necessary in all cases. However, having an awareness of these competencies and their importance is crucial for innovators to be able to identify the need for expertise in the area, which could then be outsourced.

**Table 2. Summarized Consensus Statements and IPRAS ratings.**

| # | Statement | Disagreement Index | Median | Consensus A=Agreement |
|---|---|---|---|---|
| 1 | **Preamble**: social innovation definition | 0.29 | 8 | A |
| **Mindset** | | | | |
| 2 | Developing **empathy** so that the beneficiaries' needs are understood | 0.13 | 8 | A |
| 3 | Developing **resilience** should be an important focus area for learners | 0.13 | 8 | A |
| 4 | **Adaptability** to rapidly iterate and respond to local contexts | 0.13 | 9 | A |
| **Skills: Communication** | | | | |
| 5 | **Pitching skills** are an important competency | 0.13 | 8 | A |
| 6 | **Engaging communities** is a key skill to develop | 0.1 | 9 | A |
| 7 | **Storytelling** is an important skill | 0.13 | 8 | A |
| **Intersectionality** | | | | |
| 8 | Lessons on navigating diverse cultural contexts are essential | 0.13 | 9 | A |
| 9 | Understanding **health disparities** related to intersectional issues | 0.13 | 9 | A |
| 10 | Skills to identify **social determinants** that cause health disparities | 0.13 | 9 | A |
| 11 | Prioritizing indigenous talents and wisdom is an important skill | 0.13 | 9 | A |
| 12 | Engage with community members, conduct needs assessments, and gather insights into the target population is important | 0.13 | 9 | A |
| **Evaluation** | | | | |
| 13 | Teaching **evaluation** of social innovation/impact assessment | 0.13 | 8 | A |
| **Dissemination** | | | | |
| 14 | **Manuscript writing** is a key skill to learn | 0.65 | 7 | A |
| 15 | **Effective media use** (incl. social media) for information dissemination | 0.19 | 8 | A |
| **Entrepreneurship** | | | | |
| 16 | Navigating **funding pathways** such as grant writing, securing investment and business model knowledge is an important skill | 0.29 | 8 | A |
| 17 | Lessons on building and **managing a team** are important | 0.29 | 8 | A |
| 18 | Teaching about creating sustainable innovation is important | 0.13 | 9 | A |
| 19 | Lessons in **network management** are important | 0.29 | 8 | A |
| **Ethics and Advocacy** | | | | |
| 21 | **Ethical considerations** and pathways should be included | 0.16 | 9 | A |
| 22 | **Advocacy tools** should be taught to overcome barriers in health systems | 0.13 | 9 | A |
| **Practical Tools** | | | | |
| 20 | **Crowdsourcing** to identify social innovations is a useful tool to teach | 0.23 | 8 | A |
| 23 | **User-centred design** is important to deliver comprehensive training | 0.13 | 8 | A |
| 24 | **Mentoring** is an important tool to achieve core learning competencies. | 0.29 | 8 | A |
| 25 | **Prototyping** is important in allowing innovators to test & iterate their ideas | 0.29 | 9 | A |
| 26 | Using **generative play** can be a useful tool | 0.29 | 8 | A |
| 27 | Definitions, theories, frameworks, and case studies are all key tools | 0.26 | 8 | A |
| 28 | Navigating government policy, including regulatory landscapes | 0.29 | 8 | A |
| **Research** | | | | |
| 29 | **Research skills,** including conducting literature reviews | 0.37 | 7 | A |
| 30 | Community-based participatory research | 0.13 | 9 | A |

Developing a knowledge base on intersectional issues, advocacy, ethics, and the practical tools that can be used in social innovation were all statements that also met consensus. Intersectionality refers to the complex, cumulative way in which the effects of multiple forms of discrimination (such as racism, sexism, and classism) combine, overlap, or intersect, especially in the experiences of marginalized individuals or groups [25]. Knowledge of intersectionality in this context covered topics such as understanding health disparities, considering how social determinants of health influence these disparities, and navigating diverse cultural contexts respectfully. An understanding of these topics is key for alignment with social innovation values, which are community-centricity, co-creation, and a deep respect for local context. The median Likert rating was 9, with a very low disagreement index of 0.13 on all five statements relating to intersectional themes. This demonstrates participants' agreement on the importance of learning about these issues to advance health equity through social innovation.

Knowledge of advocacy and ethics were also valuable competencies identified by the participant cohort. Commentary from several participants indicated that the inclusion of ethical frameworks and guidelines and the establishment of clear accountability measures were significant. Through the consensus-building process, advocacy tools were defined as including policy analysis and lobbying. Respondents expressed that such techniques are useful to influence decision-makers and shift their mindsets.

The final list of competencies is summarized in Fig 3, with key themes highlighting community engagement, social entrepreneurship, and knowledge of intersectionality and health disparities.

### Learning resources

As part of the second survey, contributors suggested open-access learning resources, which included a variety of tools such as videos, practical guides, free online courses, websites, and case studies. WHO/UNAIDS toolkit for community engagement, SESH grant-o-thon, Massive Open Online Courses, success stories, budget and proposal templates, and monitoring and evaluation framework were also highlighted. Repeated themes mentioned were the availability of open-access resources and the availability of material in local languages to promote health equity.

### Online training workshop

A total of 137 participants completed the exit survey. Based on the final evaluation survey, most participants were from the Western Pacific Region (50.4%, n = 69) and upper-middle-income countries (59.9%, n = 82). About 70.1% of responders (n = 96) were female, and a small number identified as belonging to a racial or ethnic minority group (10.2%, n = 14). Over half (59.9%, n = 82) were students, and nearly half (46.0%, n = 63) held a master's degree. Public health was the most common field of previous training (42.3%, n = 58), with research (45.5%, n = 25) being the primary job duty. In addition, our participants had an average of 9 ± 6.9 years of work experience. As for the post-workshop experiences, 52.6% (n = 72) of the participants attended all eight workshop sessions. The workshop goals were mostly or completely met for 78.1% (n = 107) of participants, and the majority rated the overall workshop series as excellent (67.9%, n = 93). Nearly all participants (96.4%, n = 132) expressed willingness to attend future workshops (S2 Table).

The training impact among participants was evaluated through a pre-and post-assessment of their self-reported social innovation knowledge and skills, using a scale from 1 (not at all knowledgeable) to 5 (very knowledgeable), as well as their confidence in applying these concepts post-workshop, ranging from "not confident at all" to "very confident". Additionally, participants rated their competency in seven key learning areas on a scale from 1 (not competent at all) to 5 (very competent). Overall, participants increased their social innovation knowledge scores by 1.0 and improved their confidence by 1.0 points (Fig 4). Most participants reported a significant improvement, with a 1-point increase in learning competency across six areas after completing the workshop.

 

| Icon | Theme | Description |
|------|-------|-------------|
| | Preamble | ❖ Definition of social innovation<br>❖ Purpose of study<br>❖ Intended end-users of study outputs |
| | Community Engagement | ❖ Crowdsourcing<br>❖ Community-based participatory research for the co-creation of solutions |
| | Communication | ❖ Pitching innovations for support<br>❖ Storytelling to inspire<br>❖ Engaging with communities |
| | Social Entrepreneurship | ❖ Financing/funding innovations<br>❖ Generating revenue<br>❖ Team building<br>❖ Sustainable innovation |
| | Research | ❖ Manuscript writing for dissemination<br>❖ Reviewing literature<br>❖ Ethical approval & considerations |
| | Mindset | ❖ Empathy for end users<br>❖ Resilience to adversity<br>❖ Adaptability to ambiguous circumstances |
| | Practical Tools | ❖ Mentorship for advice and feedback<br>❖ User-centred design<br>❖ Generative play in the form of games to stimulate creativity<br>❖ Prototyping to test and iterate |
| | Intersectionality | ❖ Empowering indigenous talent<br>❖ Knowledge of health disparities<br>❖ Respecting cultural norms<br>❖ Social determinants of health |
| | Advocacy | ❖ Influencing Decision Makers<br>❖ Navigating Government Policy/Regulations |

**Fig 3. Summarized Learning Competency Statements: Second Survey.**

In the context of what participants found the most useful in the workshop series, our analysis generated several themes: 'developing social innovation core competencies', 'applying knowledge for real-world impact', 'embracing diverse perspectives and building collaborations', 'engaging in interactive and practical learning', 'offering mentorship and providing guidance' 'inspiring and motivating through shared experiences'. Specifically, in the theme of 'developing social innovation core competencies', participants highlighted how they had been exposed to the concept, knowledge, and skills of social innovations in health: "*I learned a lot about social innovations in health and how they can apply to tackle health and developmental challenges in a largely populated third world country like Nigeria*"; "*Create impactful, equitable, and sustainable health solutions by actively involving communities and leveraging their strengths*". In addition, in terms of applying knowledge for real-world impact, some of our participants emphasized how they had benefitted and enhanced their capacity to address real-world challenges: "*The workshop's practical tools…. directly enhanced my projects and boosted my confidence…* "; " *Applying theories and best practices … was invaluable*".

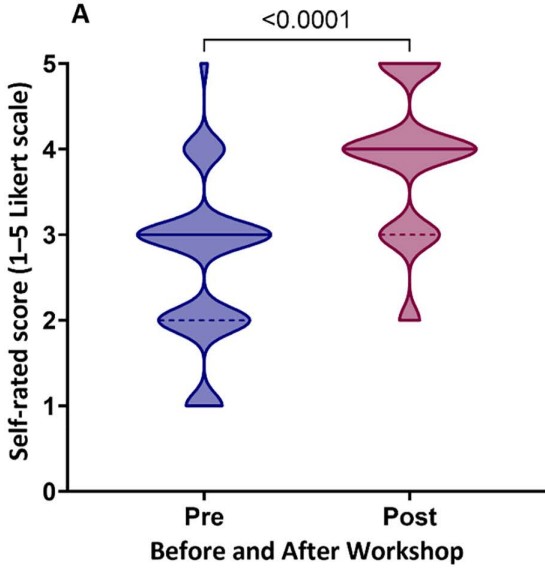

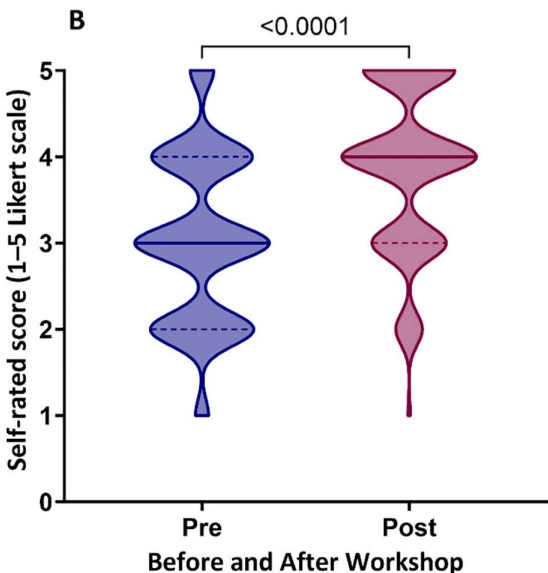

**Fig 4. Overall enhancement in participants' (A) social innovation knowledge/skills and (B) confidence in applying social innovation concepts, comparing scores before (Pre) and after (Post) the workshop.**

## Discussion

This study identified and piloted learning competencies focused on social innovation in health. The findings from the consensus process and pilot training workshops will be valuable to those learning about social innovation who aim to deliver solutions to improve health, as well as those teaching social innovation courses. This expands the literature by focusing on learning about social innovation, capturing ideas from a diverse LMIC end-user group, and leveraging participatory methods critical for learning about social innovation.

Some of the competencies with the greatest consensus included community engagement, social entrepreneurship, and knowledge of intersectional issues. Community engagement was identified to be a crucial skill. It is well recognized as one of the main pillars of social innovation in the literature and is often contained within the definition of social innovation [26]. There are several mechanisms to engage communities, including community-based participatory research and crowd-sourcing, which are recognized in this consensus statement [27].

Knowledge of intersectionality was also identified to be a core competency. There were five statements relating to inter-sectionality, all of which achieved high levels of agreement. Knowledge of intersectional issues is key to addressing health problems alongside related social determinants of health, and it has been consistently identified in the literature and open calls. Achieving an understanding of these intersectional topics is sometimes omitted from social innovation education; however, there are many teaching resources focused on intersectionality, and there is a growing body of literature linking intersectionality and social innovation [28–30].

Social entrepreneurship skills also were deemed to be important for learning about social innovation. Leveraging a net-work and collaborating with those with varied expertise and skills to build the right team are also essential skills in social entrepreneurship, and the majority of participants agreed that they were crucial [31]. There is some limited but growing literature emphasizing the importance of social entrepreneurship skills among health practitioners and communities, but additional resources are needed to help learners [32].

In terms of the learning resources participants suggested to help achieve these competencies, there was a range from online courses, conceptual frameworks and literature. These questions were included as currently many educational materials are inaccessible to learners in LMICs due to financial and infrastructural barriers. Gathering suggestions for open-access resources can help inform future resource development. Ideally, these resources could be linked to certain competencies to be used as tools to support competency-specific development. Important considerations in analysing the utility of these resources were open access status, focus on social innovation, and applicability in a range of geographies, particularly LMICs. The resources provided were all in the English language, and so diversifying these would be useful in ensuring equity of access.

One of the study's main strengths was the diverse representation. 22/46 (47.8%) of participants in the first survey were from LMICs. Several aspects of the study enabled this recruitment of such a diverse group. As with the recruitment process, both surveys and the teleconference were online, improving accessibility as participants were not required to travel. The scoping review and open call were also led by researchers from LMICs. While much of the literature that was analysed in the scoping review was produced in HICs, many of the open call submissions were from LMICs allowing for diverse experiences to be accounted for in the initial development of statements. Ultimately, this diversity will facilitate broader translation of findings.

This study also has limitations. First, the modified Delphi methodology has been criticized for lacking reliability and reproducibility. We anticipated this and increased the rigor by using participatory approaches, recruiting diverse partic-ipants, and reporting according to the ACCORD consensus reporting guidelines. Second, our participants were more familiar with social innovation compared to other people. However, this consensus statement would be most relevant for teachers, professors, and others responsible for organizing social innovation courses.

Participant dropout during the Delphi process may have influenced the consensus by reducing the diversity of per-spectives and potentially skewing results toward the views of those who remained engaged. However, steps were taken to mitigate attrition including sending reminder emails to participants 5 days before the survey completion deadline and coordinating repeat teleconference in consideration of different time zones that participants were based in. The number of participants for both surveys still ranged between 30–50 which has been considered as an optimum number for modified Delphi processes in order to allow for diverse representation [22,33].

Finally, we did not examine all core competencies in the pilot. However, we did examine 12 of the critical learning competencies identified in the Delphi study. The purpose of the workshops was to pilot the feasibility and effectiveness of

delivering the competencies through an online platform. While the pilot workshops demonstrated improved self-reported competencies, a key limitation is the absence of objective outcome measures, such as post-intervention behavioural changes, application of skills in practice, or the number of implemented projects, which restricts the ability to assess real-world impact.

Our data has implications for policy and practice. The data from this consensus statement can inform practical guides on social innovation learning. This can be widely distributed to ensure consistency in achieving social innovation learning competencies. Further evidence is needed on the best ways to support learners in achieving the outlined competencies and an evaluation on whether training translates to sustained innovation. More open-access resources should be developed and made available targeting these competencies. Furthermore, as most of the existing resources for learning about social innovation in health are in English, translation of key tools into local languages so that they are accessible to a more diverse range of communities.

To strengthen policy relevance, global health institutions (e.g., WHO, TDR) could help institutionalize social innovation in health competencies as part of training programs. To operationalize these competencies within existing training infrastructures, they could be integrated into in-service training modules for health professionals, embedded within continuing professional development programs, or included in global health and public health degree curricula. Additionally, online learning platforms and Massive Open Online Courses (MOOCs) could offer flexible, scalable delivery options. For new training programs, co-creation with local communities and stakeholders can ensure contextual relevance and support for participatory approaches. Additionally, aligning funding priorities with demonstrated capacity in these competencies could incentivize organizations, including NGOs and research teams, to adopt more participatory and locally grounded approaches.

## Conclusion

To conclude, a two-round modified Delphi was conducted, examining the core learning competencies that should be achieved by those learning about social innovation in health. Participants were recruited to represent global social innovation activity, and statements were developed based on a scoping review and international open-call findings. Consensus was achieved on all statements through a participatory approach. Mindset, skills, and knowledge in the appropriate domains contribute to success in social innovation. Innovative ways of learning should be considered to achieve important learning competencies, which may include structured content, self-directed learning, and experiential learning. The findings from this study will inform the development of a practical framework. This will be important in capability building in social innovation education to solve complex global health challenges.

## Supporting information

**S1 Table. Description of each session of the Social Innovation in Health Mid-year Training Workshop.**
(DOCX)

**S2 Table. Expression of Interest Form.**
(DOCX)

**S3 File. Survey One.**
(DOCX)

**S4 File. Survey Two.**
(DOCX)

**S5 Table. Demographic characteristics of participants attending the Social Innovation Mid-year Training Workshop held between June 2024 and July 2024 (N = 137).**
(DOCX)

**S6 Fig. Enhanced social innovation in health knowledge/skills across seven competencies: A.** Understanding health disparities and their roots in intersectional inequities: B. Building empathy and the ability to deeply connect with local communities of interest; C. Developing leadership skills to nurture relationships with local communities, enhance multi-stakeholder partnerships, and manage diverse groups; D. Cultivating a growth mindset to manage and expect failures, learn over time, and develop resiliency; E. Practicing adaptability to rapidly iterate and respond to local contexts and feedback; F. Enhancing communication skills to effectively communicate with a broad range of communities, especially people with lived experience and potential partner; G. Strengthening entrepreneurial skills to raise funds for social innovation, develop innovative financing and sustainability approaches, and rapidly iterate ideas reported by participants attending the training workshop. Participants self-rated their competency in these seven main areas following workshop completion compared to their competency before training. The violin width represents the number of participants at a certain value. Solid lines indicate the median value, and dotted lines indicate the IQR. Statistical significance between groups was assessed using the Mann–Whitney U test.
(TIF)

**S7 Table. Full Consensus Statements.**
(DOCX)

**S8 File. Supporting Manuscript: Core competencies for social innovation in health training: Evidence from a global scoping review and crowdsourcing open call.**
(PDF)

**S9 File. Social Innovation in Health Post training workshop survey.**
(DOCX)

## Acknowledgments

We would like to thank all participants in the open call.

## Author contributions

**Conceptualization:** Emily Wallace, Yusha Tao, Angelica Borbón, Joseph D. Tucker.

**Formal analysis:** Yusha Tao.

**Methodology:** Emily Wallace, Ogechukwu B. Aribodor, Zixuan Zhu, Elizabeth Chen, Joseph D. Tucker.

**Project administration:** Emily Wallace, Jana Deborah Mier-Alpano.

**Supervision:** Elizabeth Chen, Joseph D. Tucker.

**Visualization:** Yusha Tao, Beatrice Halpaap.

**Writing – original draft:** Emily Wallace, Yusha Tao, Joseph D. Tucker.

**Writing – review & editing:** Ogechukwu B. Aribodor, Zixuan Zhu, Angelica Borbón, Bertha M. Chakhame, Eunice C. Jacob, Fatema Ahmed, Joel Msafiri Francis, Komang G. Septiawan, Kovey Mawuli, Linet Mutisya, Marlita Putri Ekasari, Nwadiuto Okwuniru Azugo, Tina Fourie, Adriana S. Ruiz, Jackeline Alger, Abigail Ruth Mier, Weiming Tang, Gloria Aidoo-Frimpong, Jackline Nanono, Jesson James A. Montealto, Obidimma Ezezika, Per Kåks, Wenjie Shan, Jana Deborah Mier-Alpano, Gifty Marley, Beatrice Halpaap, Elizabeth Chen, Joseph D. Tucker.

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
