## [Decision Letter · Decision Letter 0]

24 Apr 2025

Dear Dr. Wallace,

Thank you for submitting your manuscript to PLOS ONE. After careful consideration, we feel that it has merit but does not fully meet PLOS ONE’s publication criteria as it currently stands. Therefore, we invite you to submit a revised version of the manuscript that addresses the points raised during the review process.

We look forward to receiving your revised manuscript.

Kind regards,

Monica Duarte Correia de Oliveira

Academic Editor

PLOS ONE

Journal Requirements:

“We would like to thank all participants in the open call. The work received support from TDR, the Special Programme for Research and Training in Tropical Diseases co-sponsored by UNICEF, UNDP, the World Bank and WHO. TDR is able to conduct its work thanks to the commitment and support from a variety of funders. These include our long-term core contributors from national governments and international institutions, as well as designated funding for specific projects within our current priorities. For the full list of TDR donors, please visit TDR’s  website at: https://www.who.int/tdr/about/funding/en/ TDR receives additional funding from Sida, the Swedish International Development Cooperation Agency, to support SIHI.”

“The author(s) received no specific funding for this work”

“I have read the journal's policy and the authors of this manuscript have the following competing interests:  Angelica Borbón, Bertha M. Chakhame, Adriana S. Ruiz, Abigail Ruth Mier, Obidimma Ezezika, Jana Deborah Mier-Alpano, and Gifty Marley have received honoraria for speaking at the pilot workshops. These payments were provided by SESH, the SIHI Hub in China. The authors declare no other potential conflicts of interest.”

5. We noted in your submission details that a portion of your manuscript may have been presented or published elsewhere. “The data, figures and results in this manuscript are all original and will not be published elsewhere. However, the study was informed by findings of a previously conducted literature review and open call in terms of initial survey development and workshop design. The findings of the literature and open call will be considered for publication elsewhere. “

6. Please include a copy of Table 2 which you refer to in your text on page 14.

7. Please include captions for your Supporting Information files at the end of your manuscript, and update any in-text citations to match accordingly. Please see our Supporting Information guidelines for more information: http://journals.plos.org/plosone/s/supporting-information .

**Additional Editor Comments:**

The study has merits, is within the scope of PLOS One, and of interest to the journal readers. In my view, the referees have done a very careful revision and all their comments are relevant and should be addressed by the authors when reviewing the manuscript.

Reviewers' comments:

Reviewer's Responses to Questions

**Comments to the Author**

1. Is the manuscript technically sound, and do the data support the conclusions?

Reviewer #1: Partly

Reviewer #2: Partly

2. Has the statistical analysis been performed appropriately and rigorously?

Reviewer #1: N/A

Reviewer #2: N/A

3. Have the authors made all data underlying the findings in their manuscript fully available?

Reviewer #1: Yes

Reviewer #2: Yes

4. Is the manuscript presented in an intelligible fashion and written in standard English?

Reviewer #1: Yes

Reviewer #2: Yes

Reviewer #1: 1. While the manuscript presents an innovative methdological approach, several issues identified in the review need to be addressed for the manucript to be technically sound.

2. While the manuscript presents a systhesis of results, several components (e.g. complete search strategies, full data extraction form, the complete reference list of scoping review studies) are missing or insufficiently detailed. manuscript text. This would ensure full transparency and reproducibility.

Reviewer #2: This paper addresses the research gap of lacking consensus about core competencies for social innovation learning in health. The authors used a modified Delphi process to reach consensus on core competencies and subsequently piloted them through participatory workshops. Participants were recruited in diverse parts of the world. Overall consensus about the competencies was high to very high. Competencies in the areas of community engagement, social entrepreneurship, and knowledge of intersectional issues were some of the competencies with the greatest consensus among the participants. Participants of the piloting could increase self-reported knowledge on social innovation in health.

The subject of learning for social innovation in health is highly relevant as social innovations can address today’s numerous global, regional and local health(care) challenges. Reaching consensus on how to train social innovation actors is crucial to create new and scale existing social innovations. The global focus of the study is a positive aspect as most social innovation research is done on a local or regional case study basis, which makes it difficult to transfer useful findings into wider practice. I consider the modified Delphi process paper’s an appropriate method to find consensus on core competencies for social innovation in health.

However, the paper has a number of drawbacks. First, embedding the study into the wider context of social innovation and health is missing, and relations of the study to previously published research is scarce. Second, key concepts of the paper, such as core competencies and social innovation are lacking clear definitions. Third, I do not understand the goal of the piloting and how it relates to the Delphi process. Fourth, the methodological process is difficult to follow. Fifth, the results are partly not aligned with the methodological steps. Sixth, the discussion section does not discuss all the results. In the following, I will elaborate the drawbacks in more detail.

1. Embedding the study into the wider context of social innovation and health is missing, and relations of the study to previously published research is scarce:

a. In the background section you should explain why social innovations in health are important on a broader scale. This is important to emphasise the relevance of your research: Why do communities need more effective, sustainable, and efficient solutions to serve the health needs, as you say? Why do we need learning for social innovation in health beyond communities?

b. Further, a more elaborate review of learning competencies in social innovation (in health) is needed: what competencies are mentioned by the literature (e.g. by the mentioned SI education programs?) This will help the reader get a sense of the type of competencies mentioned and their range.

2. Key concepts of the paper, such as core competencies and social innovation are lacking clear definitions.

a. Throughout the paper, I was wondering what exactly you mean by “social innovation” and “(core) competencies”. As these are key concepts of your paper, I suggest to give clear definitions. This is particularly important for social innovation as these are defined in various ways in the literature. Maybe the paper would benefit from a concepts section just after the background section where you clarify your key concepts.

3. I do not understand the goal of the piloting and how it relates to the Delphi process.

a. First, the purpose and goal of piloting the competencies and how it connects with the Delphi process is not explained in the Introduction. Then, you do not discuss the piloting in the discussion section: Please discuss the implications of your results for teaching social innovation competencies considering other studies.

b. I wonder if the paper would be better structured and focused if you focused only on the Delphi process and used the results of the competencies piloting for another paper (for example, combined with a more elaborated/enhanced impact analysis). However, of course, you decide which data you want to include in your paper.

4. The methodological process is difficult to follow

a. Please insert a reference to the "Supporting Material Literature rv and open call" manuscript in the survey development section. When I read this section I was wondering what kind of literature was reviewed, who and how many persons participated at the crowdsourcing, how you organised the crowdsourcing and how you got from the literature review and the open call to the 28 statements on core competencies (round one). Then I realised that all this information is mentioned in the Supporting Material manuscript. Even though this information is thoroughly documented in the Supporting Material manuscript, I think it would be reader-friendly to briefly address these aspects here.

b. Also in the survey development section, it sounds like the skills and knowledge-based competencies were pre-defined themes or categories you discussed in the open call. Or were these themes identified during the open call? It is mentioned in the Supporting Material manuscript but here these categories come out of the blue.

c. In the Round One section you write that the questionnaire contained 28 statements on core competencies. How did you get to those core competencies? What is the difference of these core competencies compared to the competencies derived from the scoping literature review and the crowdsourcing open call?

d. There are several unanswered questions in the Round Two section which are relevant to follow the Delphi procedure: how exactly were the statements for round two developed? In which way did you revise the statements of round one? Were the statements already revised in the teleconference session? How did you develop the additional statements? Were these the new competencies suggested in the teleconference sessions?

e. In the Round Two section you mention that participants had the opportunity to suggest useful open-access resources and provide insights on how learning content delivery could be tailored to meet the needs of resource-limited contexts. I do not understand how finding useful open-access resources and insights on the way learning content delivery could be tailored to meet the needs of resource-limited contexts was part of the goal of your research. This is a new topic that was not discussed before. I was also wondering why you asked these questions only in the second round and not already in the first round. Also, to me, it seems incoherent to ask questions about new topics in the second round of a Delphi process.

f. In the Analysis section, on line 158, you mention “item” for the first time. This term confused me: After reading on, I understood that "items" are the competencies the statements were developed for. But in figure 1, "items" seem to be the statements. So, do you use "item" and "competencies" interchangeably? I am also confused because you do not mention whether there is only one statement per competency or whether there are more than one. Am I correct in my assumption that there is one statement per competency/item, and you try to find out via the statement if there is agreement on the competencies?

g. In the Pilot Online Workshop section, you write that 12 core competencies were piloted. Please add how you chose the 12 core competencies and what these competencies were.

5. The results are partly not aligned with the methodological steps.

a. In the methods section you write that there was a discussion on statements that had conflicting comments from the first survey. What were the results of these discussions? You don't report them in the Teleconference Sessions section.

b. On line 241 you write that you identified 30 competencies, but you do not explain how you identified them. Does each of the 30 statements of the second survey correspond to on of the 30 competencies?

c. Are the Learning Resources & Project Outputs part of the results of the second survey? I find it difficult to assign the results to the different method steps. Please be more structured in reporting the results, indicating which result comes from which part of the methodological process. Perhaps a table indicating the main results for each methods step might be helpful - but take this only as a suggestion.

6. The discussion section does not discuss all results

In the discussion section you discuss the competencies with the greatest consensus. But discussion about the results of the piloting online training workshops and learning resources is missing. This is crucial to connect all parts of your study and achieve a coherent paper.

Additionally, there are some minor issues:

- On line 118 you refer to figure 1 but the correct figure is figure 2.

- Please insert a reference to table 1 in the first paragraph of the results section. The demographic details of the participants of round two should be provided to see the changes in the composition of the participants.

- In figure 1, please indicate that 34 participants joined the teleconferences sessions.

- On line 231 you write that one participant was excluded from the second survey. So, 35 participants were left but in figure 1 you indicate that 34 participants participated at the second survey.

- On line 241 you write that you identified 30 competencies but figure 2 lists less than 30 competencies.

- On line 289 you refer to table 2 but in the appendix it is table 1.

- In table 1 you mention the preamble as a theme. This is the first and only time the preamble is mentioned in the paper. I do not understand what it is. Also, the three bullet points of the preamble description are not competencies, in contrast to the bullet points in the description of the other themes.

I wish the authors good luck with revising the paper.

**Do you want your identity to be public for this peer review?** For information about this choice, including consent withdrawal, please see our Privacy Policy

Reviewer #1: No

Reviewer #2: **Yes: ** Pascal Tschumi

---

## [Author Response · Author response to Decision Letter 1]

5 Jun 2025

Dear Prof Monica Duarte Correia de Oliveira,

Thank you for the helpful comments on this manuscript. We have used the comments to strengthen the manuscript. Find below point-by-point responses.

Sincerely,

Emily and Joe on behalf of the co-authors

UPDATED FUNDING STATEMENT: The work received support from TDR, the Special Programme for Research and Training in Tropical Diseases co-sponsored by UNICEF, UNDP, the World Bank and WHO. TDR is able to conduct its work thanks to the commitment and support from a variety of funders. These include our long-term core contributors from national governments and international institutions, as well as designated funding for specific projects within our current priorities. For the full list of TDR donors, please visit TDR’s website at: https://www.who.int/tdr/about/funding/en/ TDR receives additional funding from Sida, the Swedish International Development Cooperation Agency, to support SIHI.

UPDATED COMPETING INTERESTS STATEMENT: The authors declare the following potential conflicts of interest: Angelica Borbón, Bertha M. Chakhame, Adriana S. Ruiz, Abigail Ruth Mier, Obidimma Ezezika, Jana Deborah Mier-Alpano, and Gifty Marley have received honoraria for speaking at the pilot workshops. These payments were provided by SESH, the SIHI Hub in China. They declare no other potential conflicts of interest. This does not alter our adherence to PLOS ONE policies on sharing data and materials.

COMMENT 1: The scoping review (ScR) currently included as Supplementary Material plays a fundamental role in the development the Delphi process and conceptual framework. I recommend incorporating it within the main manuscript (after addressing the specific scoping review comments identified below) to allow readers to fully understand this critical methodological component and its influence on subsequent phases of your research.

RESPONSE 1: We agree that the scoping review plays a fundamental role in the Delphi process. The scoping review and open call are detailed in a separate related paper, which is currently under peer-review with another journal. We included it as supplementary material because the pre-print was not available. We have now referenced the pre-print. This does not constitute dual publication, as the scoping review and the open call employed distinct methodologies to this paper. Additionally, it is beyond the scope of a single paper to provide a detailed account of the open call and scoping review alongside the modified Delphi process described in this manuscript.

We have also elaborated on the scoping review and open call in this paper as advised providing further details on the methodology and how it informed the Delphi process and pilot workshops.

COMMENT 2: Introduction. While the Introduction identifies gaps in social innovation education (e.g., inconsistency in competencies and resource disparities between high- and low-income settings), it would benefit from a stronger foundation on the concept of social innovation itself, its application in health contexts, and its positioning within educational frameworks. This additional context would help situate your study within the broader scholarship in these intersecting fields.

RESPONSE 2: We agree that providing additional context would benefit this paper. We have expanded the introduction section to reflect these key points.

COMMENT 3: Introduction. I encourage you to elaborate on the broader implications of standardizing competencies. While you mention the study will "inform training programmes," expanding on how this standardization could influence policy, scale innovations, or reduce health disparities (particularly in LMIC contexts) would enhance the manuscript's relevance. Consider connecting your research to potential outcomes such as workforce development and community empowerment.

RESPONSE 3: We agree that there are broader implications to standardizing competencies beyond informing training programmes. We have elaborated on this in the updated manuscript to describe the benefits of aligning educational curricula including in LMIC contexts. Ultimately, this standardization could help to empower local populations to become active participants in co-producing health outcomes which could lead to reduced health disparities.

COMMENT 4: Introduction. Your grouping of competencies into knowledge, skills, and mindset categories would benefit from theoretical justification. For instance, the KSA (Knowledge-Skills-Attitudes) framework proposed by Benjamin Bloom (Bloom et al., 1956), which is widely recognized across education, and human resource training literature, could be employed.

RESPONSE 4: We agree that justifying the grouping of competencies would strengthen the paper and we have included a reference to the KSA framework from Benjamin Bloom.

COMMENT 5: Introduction. The transition from problem identification to study objectives appears somewhat abrupt. Consider reorganizing to create a clearer connection between the identified gaps (e.g., fragmented LMIC training resources) and your study's specific aims. Framing the study as a response to WHO/TDR's call for action would enhance narrative cohesion.

RESPONSE 5: We agree that the transition was somewhat abrupt, and we have edited to ensure better flow between the identified gaps and the study’s specific aims.

COMMENT 6: Introduction. While your introduction highlights the importance of social innovation in global health education, explicit research questions would strengthen the manuscript. I recommend adding a concise paragraph stating your specific research questions, demonstrating how they align with the identified gaps, and outlining how you intend to address them.

RESPONSE 6: We agree that including explicit research questions would strengthen the manuscript hence we have included an additional paragraph at the end of the introduction to address this. It highlights that the study aims to identify the core competencies in social innovation in health and explore whether these competencies could be effectively taught and engaged with in a virtual setting.

COMMENT 7: Methods. Please clarify how the scoping review and crowdsourcing open call specifically informed the Delphi process design. An overview figure linking each methodological step would provide valuable clarity on how these components connect and build upon each other.

RESPONSE 7: We agree that elaborating on how the open call and scoping review informed the Delphi would be beneficial. To address this, we included further details on the open call and scoping review, including the number of submissions and studies included and that the themes extracted from both the open call and literature review were constructed into statements which were distributed in round one of the modified Delphi process.

COMMENT 8: Methods. The relatively low participation rate in the first Delphi round (67.6%) and further attrition in Round 2 raise potential bias concerns. I recommend discussing steps taken to mitigate attrition and acknowledging its potential impact on consensus validity.

RESPONSE 8: We agree that it is important to discuss steps which were taken to mitigate attrition and to acknowledge its potential impact. We have included further detail on this in the discussion section. We outline that participant dropout during the Delphi process may have influenced the consensus by reducing the diversity of perspectives and potentially skewing results toward the views of those who remained engaged. However, steps were taken to mitigate attrition, including sending reminder emails and coordinating multiple teleconferences.

COMMENT 9: Methods. The rationale for piloting 12 out of 30 competencies requires clearer explanation. Please revise this section to detail why these specific competencies were selected for piloting over others.

RESPOSNE 9: We agree that further clarity on why only 12/30 competencies were piloted would improve the paper. We have now described that only twelve out of thirty were piloted due to time and resource constraints as well as the areas of expertise of the facilitators. The workshops were co-designed by people who submitted the highest scoring open-call applications.

COMMENT 10: Discussion. The geographic imbalance in participant recruitment, particularly the underrepresentation of low-income regions, deserves discussion as a limitation potentially affecting the framework's universal applicability.

RESPONSE 10: Ensuring diverse representation was a priority for the research team in the recruitment of participants as well as in the study design. We have included further details on the representation from both LMICs and HICs which we believe strengthens the frameworks universal applicability. The fifth paragraph in the discussion section now describes that 22/46 (47.8%) of participants in the first survey were from LMICs. Several aspects of the study enabled this recruitment of such a diverse group. As with the recruitment process, both surveys and the teleconference were online, improving accessibility as participants were not required to travel. The scoping review and open call were also led by researchers from LMICs. While much of the literature that was analysed in the scoping review was produced in HICs, many of the open call submissions were from LMICs allowing for diverse experiences to be accounted for in the initial development of statements.

COMMENT 11: Discussion. While attrition in the Delphi process is noted, reflecting on how participant dropouts might have influenced consensus would enhance methodological transparency.

RESPONSE 11: We agree, and we have now described how participant dropout during the Delphi process may have influenced the consensus by reducing the diversity of perspectives and potentially skewing results toward the views of those who remained engaged in the discussion section. This has been revised in the discussion section.

COMMENT FROM REVIEWER 2: First, embedding the study into the wider context of social innovation and health is missing, and relations of the study to previously published research is scarce. Second, key concepts of the paper, such as core competencies and social innovation are lacking clear definitions. Third, I do not understand the goal of the piloting and how it relates to the Delphi process. Fourth, the methodological process is difficult to follow. Fifth, the results are partly not aligned with the methodological steps. Sixth, the discussion section does not discuss all the results.

RESPONSE: We agree that embedding the study into the wider context and better defining key concepts would benefit the paper and we have edited accordingly.

Regarding the third point, the goal of the piloting was to assess the feasibility and perceived value of delivering the competencies identified through the Delphi process in an online workshop format. While the Delphi process was used to reach consensus on the core competencies, the piloting phase aimed to explore whether these competencies could be effectively communicated and engaged with in a virtual setting. We have clarified this relationship in the revised manuscript.

In respect to the fourth point, we have revised figure one which depicts the methodological steps in for clarity.

Fifth, we have edited the manuscript to improve the alignment between the results and the methodological steps.

Finally, discussing each competency individually was outside the scope of this manuscript. We have highlighted the competencies which reached highest levels of consensus. We have strengthened the discussion section by elaborating on the learning resources, workshops and potential limitations.

MINOR REVISIONS

On line 118 you refer to figure 1 but the correct figure is figure 2.

RESPONSE: This is now corrected.

Please insert a reference to table 1 in the first paragraph of the results section. The demographic details of the participants of round two should be provided to see the changes in the composition of the participants.

RESPONSE: This is now included.

In figure 1, please indicate that 34 participants joined the teleconferences sessions.

RESPONSE: This is now included.

On line 231 you write that one participant was excluded from the second survey. So, 35 participants were left but in figure 1 you indicate that 34 participants participated at the second survey.

RESPONSE: This is now amended in figure 1.

On line 241 you write that you identified 30 competencies but figure 2 lists less than 30 competencies.

RESPONSE: To enhance clarity and readability, we chose not to include all 30 competencies in Figure 2, as doing so would have compromised the visual effectiveness of the graphic. Instead, we highlighted the competencies that received the highest levels of consensus and were considered most important.

On line 289 you refer to table 2 but in the appendix it is table 1.

RESPONSE: This is now corrected.

In table 1 you mention the preamble as a theme. This is the first and only time the preamble is mentioned in the paper. I do not understand what it is. Also, the three bullet points of the preamble description are not competencies, in contrast to the bullet points in the description of the other themes.

RESPONSE: We have updated the results section to give further details on the preamble. We describe that the preamble was an added item in the second survey as per feedback on the first edition. It contained the definition of social innovation, outlined the reasons for why training and education are important to achieve impactful innovation in health, the primary users of the consensus statement and different mechanisms to achieve competencies in social innovation.

Line 36: Specify the exact number of competencies piloted (12) to avoid ambiguity.

RESPONSE: We agree with this recommendation which now has been amended.

Introduction. The choice of a modified Delphi method warrants more thorough justification. I suggest explaining why this method was selected over alternatives (e.g., nominal group technique) and how it specifically aligns with your study's participatory goals. A brief discussion of the Delphi method's particular strengths in your context would strengthen methodological rigor.

RESPONSE: The beginning of the methods section now highlights why the modified Delphi was used and its advantages over other methods.

Introduction. The frequent use of institutional acronyms (SIHI, TDR, SESH) may reduce clarity for readers unfamiliar with these organizations/initiatives. Consider adding a "Study Context" section that defines each institution's mandate and relevance to your research. This would help readers understand how these partnerships enhanced your study design, recruitment approach, and policy implications.

RESPONSE: We agree, there is now a Study Context paragraph at the end of the background section.

Line 510: Renumber the table summarizing learning competency statements, as there are currently two tables designated as Table 1.

RESPONSE: Numbering of tables has now been corrected amended.

Results. A table with demographic information of the Delphi panel participants in both rounds - included in the manuscript text - would enhance methodological transparency and allow readers to assess the diversity and representativeness of your expert panel.

RESPONSE: We agree, this is now included.

Results. Figure 3 requires clarification. The y-axis scale should be clearly defined, and the caption "B. confidence in applying these concepts post-workshop" is confusing since the figure displays both pre- and post-intervention results. Consider revising for greater clarity.

RESPONSE: We agree, this is now revised

Results. Consider including the table with Delphi Round 2 results within the main manuscript rather than supplementary materials.

RESPONSE: We agree, this is now included as table 3.

Discussion. While the pilot workshops demonstrated improved self-reported competencies, I suggest acknowledging the limitation of not including objective metrics (e.g., post-intervention behavioural changes or number of projects implemented) to evaluate real-world impact.

RESPONSE: We agree, this is no

---

## [Decision Letter · Decision Letter 1]

10 Jul 2025

Dear Dr. Wallace,

Thank you for submitting your manuscript to PLOS ONE. After careful consideration, we feel that it has merit but does not fully meet PLOS ONE’s publication criteria as it currently stands. Therefore, we invite you to submit a revised version of the manuscript that addresses the points raised during the review process.

We look forward to receiving your revised manuscript.

Kind regards,

Monica Duarte Correia de Oliveira

Academic Editor

PLOS ONE

Reviewers' comments:

Reviewer's Responses to Questions

**Comments to the Author**

Reviewer #1: (No Response)

Reviewer #2: All comments have been addressed

2. Is the manuscript technically sound, and do the data support the conclusions?

Reviewer #1: Partly

Reviewer #2: Yes

3. Has the statistical analysis been performed appropriately and rigorously?

Reviewer #1: Yes

Reviewer #2: N/A

4. Have the authors made all data underlying the findings in their manuscript fully available?

Reviewer #1: Yes

Reviewer #2: Yes

5. Is the manuscript presented in an intelligible fashion and written in standard English?

Reviewer #1: Yes

Reviewer #2: Yes

Reviewer #1: (No Response)

Reviewer #2: Dear authors, thank you very much for revising your manuscript. Your paper has improved a lot. These are the major improvements based on my comments:

- The background section now provides important features of social innovation in health that were not included before. Readers can have a better idea of what social innovation in health is and why it is important.

The aim of the piloting is now more clearly formulated when you mention the aim of the study.

- After re-reading, the reference to figure 2 at the end of the Survey Development section does not fit in my opinion. I suggest no reference to any figure here.

- The methodological process is now much better explained, particularly the additional information on survey round two is very helpful to follow the methodology.

- Thanks to the additions in the Methods and Results sections it is now easier to align the methodological steps with the results.

- The discussion section now relates to all parts of your study.

There are still some minor remaining issues:

- Please insert references for the statements in the second and third sentences of the Background section.

- The Study Context section ends with “TDR” -> perhaps you intended to mention the TDR as collaboration partner?

- In the Discussion section, on line 471, you write “There are several strengths to this study including diverse representation”. Following this topic sentence, the readers expect several strengths being discussed, however you only discuss diverse representation. I suggest to label diverse representation as the main strength of your paper (if you agree, of course) and say why it is of such high relevance to have diverse representation.

I wish you good luck with publishing your paper!

**Do you want your identity to be public for this peer review?** For information about this choice, including consent withdrawal, please see our Privacy Policy

Reviewer #1: No

Reviewer #2: **Yes: ** Pascal Tschumi

---

## [Author Response · Author response to Decision Letter 2]

19 Aug 2025

Thank you for the helpful comments on this manuscript. We have used the comments to strengthen the manuscript. Find below point-by-point responses.

Sincerely,

Emily and Joe on behalf of the co-authors

COMMENT 1: Abstract. I would encourage you to improve the abstract by incorporating key specific (quantitative) results that demonstrate the strength of your findings. For instance, rather than the general statement "Some competencies reached higher levels of agreement than others" (lines-27-28) consider providing concrete percentages or numbers of competencies that achieved consensus. This specificity would better communicate the robustness of your consensus-building process to readers and enhance the abstract's overall quality.

RESPONSE 1: We agree that including quantitative results strengthens the abstract and have amended the manuscript accordingly to reflect this.

COMMENT 2: Abstract. The statement regarding "(…) most participants reported a significant improvement across six competencies." would benefit from greater precision and transparency. I recommend clarifying the measurement instrument employed. Moreover, what constituted “significance”? This additional detail would strengthen the credibility of your findings and assist readers in better understanding on how workshop participants’ improved their knowledge and skills.

RESPONSE 2: We have expanded on the statement regarding post-workshop self-reported competency among participants as much as possible within the confines of the abstract word count limitation.

COMMENT 3: Abstract. While the final sentences of your abstract are appropriate, I believe it somewhat understates the significant contributions of your work. I would suggest more explicitly articulating the practical implications for curriculum and training program design, competency framework development, and policy implementation. This enhancement would better reflect the substantial value your research offers to the global health education community.

RESPONSE 3: We agree and have elaborated on this briefly to stay within the abstract word count limit.

COMMENT 4: Introduction. In lines 75 and subsequent statements about social innovation being "underrepresented in formal educational frameworks" and training programs lacking "clarity around core competencies," I recommend strengthening these assertions with robust literature support. Additional references would substantiate these important claims and provide readers with a stronger foundation for understanding the research gap your study addresses.

RESPONSE 4: Additional references have been added to support these statements.

COMMENT 5: Introduction. To enhance the manuscript's clarity and focus, I encourage you to include a dedicated paragraph that explicitly states your specific research questions. This addition would demonstrate how your objectives align with the identified gaps in the literature and clearly outline your intended approach to addressing these important issues.

RESPONSE 5: The research questions are now explicitly highlighted in the final paragraph of the background section.

COMMENT 6: Regarding the statement on lines 233-234 about piloting twelve of thirty competencies due to "time and resource constraints as well as the areas of expertise of the facilitators," it would be valuable to have additional clarification about the facilitators' specific areas of expertise, the number of facilitators involved, and a more detailed description of the workshop procedures. This information would provide valuable context for understanding the workshop implementation approach.

RESPONSE 6: Thank you for this helpful suggestion. We have revised the Methods section to provide additional clarification on the structure and delivery of the workshop. The training series was led by finalists from a global open call, each of whom was paired with a mentor with relevant expertise in social innovation, implementation research, or community engagement. These mentors supported the finalists in reviewing and refining their training materials and co-presenting the sessions. We have also added a more detailed description of the workshop procedures.

COMMENT 7: Methods. The workshop evaluation would be significantly strengthened by including more comprehensive details about the pre-post assessment instruments and their validation procedures. I recommend considering the inclusion of these guides/templates used in the assessment surveys as supplementary material, which would enhance methodological transparency and facilitate replication by other researchers.

RESPONSE 7: We have now added more comprehensive details about the pre-post assessment instruments, including descriptions of their development and any relevant validation procedures, in the revised Methods section. Additionally, we have included the survey used in the assessment as supplementary material.

COMMENT 8: Discussion. Given the important WHO/TDR context of this work, I had anticipated a more detailed discussion of how these competencies could be operationalized within both existing training infrastructures and newly developed programs. Expanding this discussion would provide valuable guidance for practitioners and policymakers seeking to implement your framework and would enhance the practical utility of your research.

RESPONSE 8: Further details on the training infrastructures that these competencies may be operationalized in is now included in the discussion.

COMMENT 9: After re-reading, the reference to figure 2 at the end of the Survey Development section does not fit in my opinion. I suggest no reference to any figure here.

RESPONSE 9: This reference is now removed.

COMMENT 10: Line 32: A spacing correction is needed between the words "twelve" and "learning" to ensure proper formatting.

RESPONSE 10: This has been corrected.

COMMENT 11: References. I noticed some inconsistencies in the citation format throughout the reference list. Please ensure uniform adherence to the journal's citation style guidelines to maintain professional presentation standards.

RESPONSE 11: References have now been updated.

COMMENT 12: Please insert references for the statements in the second and third sentences of the Background section.

RESPONSE 12: References are now added.

COMMENT 13: The Study Context section ends with “TDR” -> perhaps you intended to mention the TDR as collaboration partner?

RESPONSE 13: This was an error and has now been corrected.

---

## [Decision Letter · Decision Letter 2]

1 Sep 2025

Dear Dr. Wallace,

Thank you for submitting your manuscript to PLOS ONE. After careful consideration, we feel that it has merit but does not fully meet PLOS ONE’s publication criteria as it currently stands. Therefore, we invite you to submit a revised version of the manuscript that addresses the points raised during the review process.

We look forward to receiving your revised manuscript.

Kind regards,

Monica Duarte Correia de Oliveira

Academic Editor

PLOS ONE

Journal Requirements:

Reviewer #1: See pdf in the platform.

Reviewer #2: Thank you very much for re-revising your manuscript. You addressed all my comments adequately, except for one:

- In the Discussion section, on line 457, you write “There are several strengths to this study including diverse representation”. Following this topic sentence, the readers expect several strengths being discussed, however you only discuss diverse representation. I suggest to label diverse representation as the main strength of your paper (if you agree, of course) and say why it is of such high relevance to have diverse representation.

I wish you good luck with publishing your paper!

Reviewers' comments:

Reviewer's Responses to Questions

**Comments to the Author**

Reviewer #1: (No Response)

Reviewer #2: (No Response)

2. Is the manuscript technically sound, and do the data support the conclusions?

Reviewer #1: Yes

Reviewer #2: Yes

3. Has the statistical analysis been performed appropriately and rigorously?

Reviewer #1: Yes

Reviewer #2: N/A

4. Have the authors made all data underlying the findings in their manuscript fully available?

Reviewer #1: Yes

Reviewer #2: Yes

5. Is the manuscript presented in an intelligible fashion and written in standard English?

Reviewer #1: Yes

Reviewer #2: Yes

Reviewer #1: (No Response)

Reviewer #2: 

Thank you very much for re-revising your manuscript. You addressed all my comments adequately, except for one:

- In the Discussion section, on line 457, you write “There are several strengths to this study including diverse representation”. Following this topic sentence, the readers expect several strengths being discussed, however you only discuss diverse representation. I suggest to label diverse representation as the main strength of your paper (if you agree, of course) and say why it is of such high relevance to have diverse representation.

I wish you good luck with publishing your paper!

**Do you want your identity to be public for this peer review?** For information about this choice, including consent withdrawal, please see our Privacy Policy

Reviewer #1: No

Reviewer #2: **Yes: ** Pascal Tschumi

---

## [Author Response · Author response to Decision Letter 3]

22 Nov 2025

Thank you for you comments. We have now addressed the final comments.

We have highlighted diversity as the primary strength of the study which will help ensure that results are broadly translatable. Additionally we have specified particular competency domains in the abstract and distinguished the conclusion section from the discussion section.

Table 3 is now appropriately referenced

---

## [Editor Report · Decision Letter 3]

7 Dec 2025

Global learning opportunities within social innovation in health (GLOWS): A modified delphi process to identify and pilot core competencies for learning

PONE-D-25-16004R3

Dear Dr. Wallace,

We’re pleased to inform you that your manuscript has been judged scientifically suitable for publication and will be formally accepted for publication once it meets all outstanding technical requirements.

Kind regards,

Monica Duarte Correia de Oliveira

Academic Editor

PLOS One

---

## [Editor Report · Acceptance letter]

PONE-D-25-16004R3

PLOS One

Dear Dr. Wallace,

I'm pleased to inform you that your manuscript has been deemed suitable for publication in PLOS One. Congratulations! Your manuscript is now being handed over to our production team.

Kind regards,

on behalf of

Professor Monica Duarte Correia de Oliveira

Academic Editor

PLOS One